# Translators as mediators to mend the psychological gap between source text and target text: A corpus-based study on the Chinese English translation of modal verbs in the Chinese Report on the Work of the Government (2000–2022)

Xujun Tian⬚*

School of Foreign Languages, Shanghai Lixin University of Accounting and Finance, Shanghai, China

* 20230010@lixin.edu.cn

## Abstract

This study examines the role of translators as mediators in bridging the psychological gap between source text (ST) and target text (TT) through a corpus-based analysis of Chinese English translation of modal verbs in the Chinese Report on the Work of the Government from 2000 to 2022. The research reveals that translators frequently modulate high-value Chinese modals into medium or low-value English equivalents. This strategic modulation, along with the use of explicitation and implicitation, reflects the translators' efforts to balance source text fidelity with target text acceptability. The findings underscore the translators' role in adapting authoritative Chinese political discourse to align with expectations of the international audiences, thereby facilitating effective cross-cultural communication. This study contributes to translation studies by providing insights into the translation of modality in political texts and emphasizing the critical role of translators in mediating linguistic and cultural differences.

## Introduction

The field of translation studies has evolved significantly over the past few decades, emphasizing not only linguistic fidelity but also the nuances of psychological distance between the source text (ST) and target text (TT) [1]. This shift in focus recognizes the translator's dual role as both a linguistic converter and a mediator of meaning [2]. Particularly, in political and official discourse, such as the Chinese Report on the Work of the Government (here in after as RWG), the translator's role becomes even more critical. As suggested by Hu and Tian [3], such documents are not merely informative texts but are vehicles for conveying policies, intentions, and national ethos. As argued by House [4], it is a challenge that requires both skill and deep understanding to ensure that the translation resonates with the target audience while maintaining the original context.

**Data availability statement:** All relevant data are within the paper and its Supporting Information files.

**Funding:** The author(s) received no specific funding for this work.

**Competing interests:** The authors have declared that no competing interests exist.

Modal verbs, which express necessity, possibility, permission, and obligation, play a crucial role in shaping the tone and intention of official documents [5]. In the context of Chinese-to-English (C-E) translation, the accurate rendering of these modal verbs is essential to preserve the document's intended meaning and effect [6]. Given the significant political and cultural differences between China and English-speaking countries, Baker suggests that these translations require more than direct linguistic correspondence; they necessitate a nuanced approach that bridges the psychological gap between the two languages and their respective audiences [7]. This study explores the way how translators function as mediators through examination of the Chinese English translation of modal verbs in Chinese RWGs from 2000 to 2022.

This research aims to investigate how the translators of the Chinese RWGs, through the adoption of linguistic choices and translation strategies, render the Chinese modal verbs into varied modal verbs and structures, with shifts in modal value, to navigate and mitigate the psychological gap between the ST and TT and explore the factors behind such practices. Psychological gap or psychological distance in translation, in this research, refers to the perceived disparity between the source text (ST) and target text (TT) caused by linguistic, cultural, and ideological differences. It reflects the challenge of ensuring that the translated text resonates with the target audience while maintaining the intent, tone, and authority of the original. Translators bridge this gap or distance by employing various translation strategies to balance fidelity to the source text with acceptability for the target audience.

## Literature review

The role of translators as mediators in bridging linguistic and cultural gaps has been a subject of increasing interest in translation studies. The concept of psychological distance in translation has been explored by several scholars. Nida introduced the idea of "dynamic equivalence," emphasizing the importance of eliciting the same response in the target audience as the source text did in its original context [1]. Building on this, Venuti discussed the "invisibility" of translators and how their choices can affect the reader's perception of cultural distance [2]. More recently, House proposed a model for translation quality assessment that considers both overt and covert translation, highlighting the role of cultural filter in managing cross-cultural differences [4].

The view of translators as cultural mediators has gained prominence in recent years. Venuti proposes that translators serve as cultural mediators who navigate between the source and target cultures to produce translations that resonate with the target audience while preserving the essence of the source text [2]. This mediator role is especially crucial in political and official documents where the stakes are high, and the potential for miscommunication is significant. Translators' adeptness at bridging cultural and psychological gaps is essential for ensuring that translated texts are both accurate and impactful [7]. Katan argues that translators should act as cultural mediators, actively intervening in the communication process to bridge gaps in understanding [8]. This perspective is particularly relevant in the context of political translations, where nuanced understanding of both cultures is crucial [9]. Recent research by Li and Pan has further emphasized the translator's role as a cross-cultural mediator in political discourse, particularly in the context of China's international communication [10].

The Chinese RWGs represent a specific genre with distinct linguistic features and translation requirements. As Lu points out, the translation of RWGs is "a serious political task," [11, p, 48] that demands both high fidelity and high effectiveness. Wu emphasizes that compared to other genres, "political text translation requires greater faithfulness to the original text," [12, p, 64] and Cheng notes that "the standards for faithfulness are much

stricter"[13, p, 18]. Feng analyzed the rhetorical structures of Chinese RGWs, noting their characteristic use of modal verbs to express policy intentions and commitments [14]. More recent studies have focused on the evolution of translation strategies in RWGs. Zhang and Liu conducted a diachronic study of translation strategies in Chinese RWGs from 2000 to 2020, revealing a gradual shift towards more reader-friendly approaches while maintaining political accuracy [15]. Hu and Afzaal further developed this line of research by establishing a multilingual corpus-based platform for political text translation, demonstrating how corpus-based approaches can enhance translation quality and consistency in government documents [16].

Recent research has also explored new dimensions in the translation of RWGs. Chen and Wang examined the role of translation memory systems in maintaining consistency in government document translation [17], while Yang and Li investigated the impact of cultural-specific elements on translation strategies in Chinese political discourse [18]. Wei conducted a comparative study of modal verb translations in Chinese and English political texts, providing insights into cross-linguistic variations in modality expression [19]. Liu and Afzaal contributed to this field by examining syntactic complexity in translated texts, revealing how simplification strategies can enhance the accessibility of translated political documents while maintaining their essential meaning [20]. Furthermore, Sun and Zhang explored the application of corpus linguistics in analyzing translation patterns in RWGs, offering new methodological approaches for studying political translation [21].

The translation of modal verbs in Chinese RGWs has received particular attention from scholars. Li and Hu observed that the English translations of RGWs employ medium and high-value modal verbs and positive semantic prosody to reflect the government's attitude towards future work arrangements and determination for implementation [22]. However, Hu and Tian suggested reducing the use of high-value modal verbs in favor of medium and low-value ones to make the translations more flexible, composed, and accessible to international audiences [3]. Wang found that the English translations of the 2015 and 2016 reports adopted a more moderate tone compared to those of 2005 and 2006 [23]. This trend towards moderation in modal verb translation has been further confirmed by recent corpus-based studies. Tian analyzes the translator's adoption of the "explicitation" strategies and the mitigation of modal verb strength in C-E translations, highlighting how macro-level norms, linguistic, and cultural differences shape operational norms [24]. Afzaal and Du analyzed syntactic complexity in translated discourse, providing valuable insights into how translators navigate between maintaining authenticity and ensuring accessibility in specialized texts [25]. Liu et al. analyzed the translation patterns of modal verbs in Chinese political texts and found a systematic shift towards more nuanced modal expressions in English translations [26].

The existing studies have undoubtedly contributed to the research on the translation of Chinese RWGs at lexical, syntactic and textual levels from the linguistic, translational and cultural perspectives. However, little attention has paid to the specific role of translators in mediating psychological distance through the translation of modal verbs in the Chinese RWGs. As such, this study, combining insights from translation studies, linguistic analysis, and corpus-based methodologies, addresses the following research questions:

1. What are the patterns and trends in the English translations of Chinese modal verbs in the Chinese Reports on the Work of the Government (2000–2022)?

2. What translation strategies and techniques have been employed in rendering Chinese modal verbs into English, and how have these strategies altered the modal value of the translations over time?

3. How do the changes in the modal value of translated Chinese modal verbs influence the psychological distance perceived by the target audience?

4. What are the underlying motivations for translators acting as mediators in bridging the psychological gap between the source text and the target text?

By addressing these questions, this study aims to contribute to the field of translation studies, particularly in the subfield of political translation and to provide practical insights for translators and policymakers alike to ensure that the translated documents are both accurate and impactful.

## Research design

### About the corpus

This study utilizes the Chinese English Parallel Corpus of Chinese RWGs 2000-2022 for investigation. This corpus comprises 23 Chinese source texts and their corresponding English translations spanning from 2000 to 2022, including two sub-corpora: the Chinese Corpus of 2000-2022 RWGs and the English Corpus of 2000-2022 RWGs. The Chinese sub-corpus contains 8,357 types and 204,827 tokens, and the English sub-corpus includes 7,907 types and 332,121 tokens, with a total corpus size of 536,948 tokens.

### Theoretical foundation

This study adopts Halliday's classification of English modal verbs [27] and Jiang and Yang's classification of Chinese modal verbs [28] as the theoretical foundation. Halliday assigns values to English modal verbs, categorizing them into high, medium, and low-value modal verbs. High-value ones include "must," "ought to," "need," "has to," and "is to," while medium-value ones are "will," "would," "shall," and "should." Low-value ones comprise "may," "might," "can," and "could" [27]. Similarly, Jiang and Yang, as well as Xu have assigned values to Chinese modal verbs, grouping them into corresponding categories [28,29]. According to their classification, "Yinggai" (应该), "Yao" (要), and "Bixu" (必须) are high-value modal verbs; "Dagai" (大概), "Jiang" (将), "Hui" (会), "Xiang" (想), and "Yinggai" (应该) fall into the medium-value group; and "Keneng" (可能), "Yexu" (也许), "Yuanyi" (愿意), and "Keyi" (可以) are in the low-value category [28,29].

The application of Halliday's modal value system [27] has been extensively explored in translation studies. Thompson utilized Halliday's framework to examine the use of modal verbs in academic writing, highlighting how modal values reflect varying degrees of authorial commitment and certainty [30]. Similarly, Downing applied Halliday's model to study modal verbs in political discourse, demonstrating how high-value modals such as "must" and "ought to" are strategically employed to assert authority and urgency, while medium- and low-value modals like "should" and "might" are used to introduce flexibility and mitigate directness [31]. In the field of translation studies, Li and Zhang further expanded this line of inquiry by analyzing modal expressions in Chinese political speeches and their English translations, demonstrating how translators navigate between different modal systems to achieve functional equivalence [32]. Baker explored the role of modality in cross-cultural communication, using Halliday's categorization to analyze shifts in modal values during the translation of English political texts into Arabic [33]. Li and Pan employed Halliday's framework to examine how modal choices reflect power relations and ideological positions in political translation [10], while Liu et al. investigated the correlation between modal value shifts and diplomatic strategies in government document translation [26].

Similarly, research applying Jiang and Yang's and Xu's classification of Chinese modal verbs [28,29] has primarily focused on cross-linguistic comparisons and translation studies. Liu and Zhang applied Jiang and Yang's framework to analyze the use of modal verbs in Chinese legal texts, identifying how high-value modals like "必须(bixu)" and "要(yao)" convey strong obligations and legal mandates [34]. Xu's corpus-based study further expanded on this by examining modal verbs in Chinese academic writing, revealing a preference for medium-value modals such as "会(hui)" and "将(jiang)" to balance assertiveness with politeness [29]. In translation studies, Chen and Wang investigated the adaptation of Chinese modal verbs into English, using Jiang and Yang's classification to highlight shifts in modal intensity that align with target language norms [35]. These studies demonstrate the applicability of Jiang and Yang's and Xu's classifications in analyzing the pragmatic and cultural dimensions of modality, particularly in contexts where linguistic and cultural mediation is required.

These studies provide a robust theoretical basis for investigating how modal verbs function as tools for expressing obligation, possibility, and necessity in translation contexts. Drawing on their classifications of English and Chinese modal verbs, this study adopts a corpus-based methodology to investigate the English translations of modal verbs in the Chinese RWGs 2000-2022, and how the translators use the translation of the modal verbs as tools to mend the psychological gap between the Chinese ST and the English TT.

## Research methodology and procedures

This study employs a corpus-based methodology to analyze the Chinese-to-English (C-E) translation of modal verbs in the 2000-2022 RWGs. The corpus-based approach offers significant advantages in translation studies. Firstly, it allows for the systematic analysis of large volumes of text, enabling researchers to uncover patterns and trends in translation strategies [36]. Secondly, this method provides empirical data support, making it particularly suited for studying specific linguistic features, such as the translation of modal verbs [26]. Additionally, corpus-based approaches facilitate comparative analyses across languages and genres, helping researchers understand how translators mediate between linguistic and cultural differences, thereby enhancing the effectiveness of cross-cultural communication [37]. These features highlight the indispensable role of corpus-based methods in advancing research in translation studies.

The research comprises several steps: First, the investigation of C-E translation of modal verbs: the modal verbs in Chinese RWGs were used as node words to extract parallel concordance lines from the Chinese-English Parallel Corpus of Reports on the Work of the Government 2000–2022, and the corresponding English translations of the Chinese modal verbs were identified, and the frequencies and patterns of these translations over the years were analyzed. Second, the classification of modal verbs by value: the values of Chinese modal verbs with their English translations were compared to determine if there is any change in modal value during the translation process, and more significantly if there is any evidence of the translators' mediation between the ST and TT. Third, year-by-year detailed analysis: a detailed examination of the English translations of "Yao" year-by-year were conducted to further investigate how the translators use different methods to mend the psychological gap between the ST and TT. Last, the factors behind the translators' shifts in the Chinese English translation of modal verbs in the Chinese RWGs were explored.

By doing so, this study aims to highlight the mediating role of translators in maintaining the psychological and cultural balance between the source and target texts by providing a thorough and systematic examination of the translation strategies used in rendering Chinese modal verbs into English.

## Findings

### C-E translation of modal verbs in 2000–2022 RWGs

This study investigates the translation of modal verbs in Chinese RWGs from 2000 to 2022, focusing on the translator's role as a mediator in bridging the psychological gap between the Chinese ST and the English TT. The corpus-based analysis reveals several significant findings that illuminate the translator's strategic choices in rendering modal verbs from Chinese to English, to mediate the psychological distance between the Chinese source texts and their English translations.

To start the investigation, the Chinese modal verbs and their corresponding English translations were identified from the corpus and made into Table 1 for further analysis.

A prominent feature was noted at the examination the C-E translation of the Chinese modal verbs that translators employ a wide variety of English modal verbs and structures to convey the nuances of the Chinese original modal verbs. As it can be seen from Table 1, almost all the Chinese modal verbs were translated into different words, phrases or structures. Second, the modal verb "Yao" stands out as it shows a clear dominance, accounting for 73.53% (1 717 occurrences) of all modal verbs (2 335 occurrences) used, while other modal verbs including "Bixu (Xu)", "Nenggou (Neng)", "Jiang", "Hui", "Keyi", "Yinggai (Ying, Yingdang)", and "Yuanyi (Yuan)", take up 10.02%, 5.82%, 5.52%, 1.37%,.1.28% and 1.11% respectively. As "Yao" indicates high degree of obligation and necessity, such prevalence of it in the Chinese RWGs suggests a strong emphasis on obligation and necessity in the original Chinese texts, reflecting the authoritative nature of government reports [38].

The translators' such practices are strategies, in Klaudy's terms [39], explicitation and implicitation. The translator employs the explicitation strategy, as is observed in most cases, and chooses several specific English modal verbs to clarify the "general, implicit or vague meaning" [37, p. 83]. This approach helps to bridge potential gaps in understanding between the source and target cultures, ensuring that the intended message is accurately conveyed to the English-speaking audience. The translators adopted the implicitation strategy, on the other hand, and omitted "Yao" in the English translation. The adoption of such strategies in different contexts suggests that the translator judges the modal meaning to be sufficiently

Table 1. English Translations of Chinese Modal Verbs in 2000-2022 RWGs.

| Node Words | Freq. | English Translation & Frequency |
|---|---|---|
| *Yao* | 1 717 | will 620, must 354, need 329, should 308, ellipsis 53, Others* 41, be to 11, can 1 |
| *Bixu (Xu)* | 234 | must 189, ellipsis 19, need 9, will 6, should 3, necessary 2, have to 1, essential 1, require 1, imperative 1, demand 1, vital 1 |
| *Nenggou (Neng)* | 136 | can 39, will 33, ellipsis 24, must 19, able to 10, capable 4, enable 3, should 1, possible 1, have the ability to 1, certain to 1 |
| *Jiang* | 129 | will 117, ellipsis11, should 1 |
| *Hui* | 32 | will 20, ellipsis 5, can 2, could 2, would 1, going to 1, able to 1 |
| *Keyi* | 30 | ellipsis 12, can 10, will 4, may 4 |
| *Yinggai (Ying, Yingdang)* | 30 | should 19, must 5, ellipsis 5, will 1 |
| *Yuanyi(Yuan)* | 26 | ready to 15, willing to 4, will 3, work 2, want to 1, ellipsis 1 |
| *Xiang* | 1 | want to 1 |

Note: Others (41) include aim at 3, designed 2, essential 4, expect to 1, going to 2, have an obligation to 1, have to 4, imperative 7, important/ of importance 6, necessary 3, vital to 2, obligated 1, slated to 1, wish to 1, would like to 1, resolved to 1, committed to 1.

conveyed by the context in these cases, avoiding redundancy and maintaining natural expression in the target language [7]. It is argued that this diversity in translation choices indicates the translators carefully selected the target language expressions that best capture the intended meaning and tone of the source text while considering the expectations and norms of the target audience, which underlines the translators' role as a mediator [40].

In addition, due to the prominence of "Yao," with the most frequencies (1 717) and being the only modal verb appeared in all the 23 Chinese source texts, its English translations were identified and counted in a yearly manner and made into Table 2 for further scrutiny.

Table 2 shows that "Yao" was translated into several different English modal verbs, phrases and structures, including *will*, *must*, *should*, *need*, *ellipsis*, *be adj to*, *be to*, and *others* in all the 23 target texts. Close examination shows that it was predominantly translated into four English modal verbs: *will* (36.11%), *must* (20.62%), *need* (19.16%), and *should* (17.94%). Such distribution reveals the translators adapted the English translation of "Yao" to suit different contexts and rhetorical purposes in the target language, which reflects the translator's nuanced interpretation of the source text's modal force. Such adaptations of the translation of Chinese modal verbs to meet the target audience and target culture can serve as a particularly insightful example of the translators' mediating function.

**Table 2.  Summary of the English Translations of "*Yao*" in 2000–2022 RWGs.**

| Year | Total freq. | will | must | need | should | ellipsis | be adj. to | be to | have to | Others |
|---|---|---|---|---|---|---|---|---|---|---|
| **2000** | 111 | 6 | 19 | 6 | 73 | 1 | 5 | 0 | 1 | 0 |
| **2001** | 77 | 3 | 14 | 30 | 21 | 5 | 0 | 1 | 2 | 1 |
| **2002** | 75 | 12 | 14 | 23 | 22 | 2 | 2 | 0 | 0 | 0 |
| **2003** | 26 | 0 | 6 | 0 | 17 | 2 | 1 | 0 | 0 | 0 |
| **2004** | 111 | 31 | 39 | 16 | 19 | 3 | 3 | 0 | 0 | 0 |
| **2005** | 85 | 29 | 20 | 23 | 9 | 2 | 1 | 0 | 1 | 0 |
| **2006** | 100 | 28 | 15 | 44 | 7 | 4 | 0 | 2 | 0 | 0 |
| **2007** | 91 | 30 | 17 | 25 | 6 | 6 | 4 | 3 | 0 | 0 |
| **2008** | 97 | 47 | 25 | 16 | 3 | 0 | 3 | 3 | 0 | 0 |
| **2009** | 77 | 46 | 5 | 23 | 2 | 0 | 1 | 0 | 0 | 0 |
| **2010** | 95 | 56 | 9 | 24 | 3 | 1 | 1 | 0 | 0 | 1 |
| **2011** | 54 | 20 | 7 | 20 | 0 | 4 | 3 | 0 | 0 | 0 |
| **2012** | 45 | 30 | 6 | 2 | 5 | 1 | 0 | 0 | 0 | 1 |
| **2013** | 41 | 0 | 0 | 7 | 32 | 0 | 1 | 1 | 0 | 0 |
| **2014** | 71 | 33 | 12 | 13 | 13 | 0 | 0 | 0 | 0 | 0 |
| **2015** | 84 | 26 | 35 | 18 | 5 | 0 | 0 | 0 | 0 | 0 |
| **2016** | 58 | 30 | 11 | 5 | 7 | 4 | 0 | 0 | 0 | 1 |
| **2017** | 98 | 40 | 19 | 11 | 22 | 2 | 4 | 0 | 0 | 0 |
| **2018** | 71 | 41 | 15 | 8 | 3 | 2 | 0 | 1 | 0 | 1 |
| **2019** | 90 | 45 | 25 | 8 | 8 | 4 | 0 | 0 | 0 | 0 |
| **2020** | 60 | 28 | 16 | 4 | 2 | 9 | 0 | 0 | 0 | 1 |
| **2021** | 34 | 13 | 8 | 2 | 8 | 1 | 2 | 0 | 0 | 0 |
| **2022** | 66 | 26 | 17 | 1 | 21 | 0 | 0 | 0 | 0 | 1 |
| **Total** | 1 717 | 620 | 354 | 329 | 308 | 53 | 31 | 11 | 4 | 7 |
| **Percentage** | 100 | 36.11 | 20.62 | 19.16 | 17.94 | 3.09 | 1.81 | 0.64 | 0.23 | 0.41 |

Note: the 6 items included in the type **Others** are: of importance (2001), would like to (2010), wish to (2012), have an obligation to (2016), expect to (2018), can (2020), and remain (2022).

These findings suggest that the translator, acting as a mediator, employs a range of strategies to navigate the psychological gap between the ST and TT. By carefully selecting appropriate modal expressions and adjusting the explicitness of the message, the translator strives to achieve a balance between fidelity to the source text's intent and adherence to target language norms and reader expectations. The above analysis supports the view that translation, particularly in the context of government documents, is not merely a linguistic transfer but a complex process of intercultural mediation [8]. The translator's choices in rendering modal verbs reflect a deep understanding of both source and target cultures, as well as an awareness of the diplomatic and political implications of language use in international communication.

## Modal value shifts in English translation of modal verbs in 2000–2022 RWGs

This section examines the modal values of Chinese modal verbs and their English translations in the Chinese RWGs from 2000 to 2022, focusing on how translators mediate the psychological gap between the ST and TT through modal value shifts.

The Chinese modal verbs and their English translations were grouped together into three categories, high, medium and low value ones and made into Table 3 for further analysis.

The comparison of the values of Chinese modal verbs with their English translations reveals several distinctive features. First, detailed analysis shows a significant discrepancy between the modal values in the Chinese texts and their English translations, as shown in Table 3. In the Chinese RWGs, high-value modal verbs dominate, accounting for 84.84% of all occurrences, while medium and low-value modal verbs comprise only 6.94% and 8.22%, respectively. However, the English translations show a markedly different distribution: medium-value modal verbs are most prevalent (48.65%), followed by high-value (39.44%) and low-value (11.9%) modal verbs. This shift in modal value distribution indicates strategic choices made by translators, consciously or subconsciously, to modulate the tone and intensity of the original text. Such modulation can be seen as an effort to bridge the psychological gap between the ST and TT, making the content more accessible and acceptable to the target audience [7].

Second, a close examination reveals that 45% of high-value Chinese modal verbs are translated into medium or low-value English modal verbs, as shown in Table 3. This downward shift in modal value is particularly evident in the translation of "Yao", a high-value Chinese modal verb that appears the most times in the corpus. In this regard, we take "Yao" as an example to examine the value shifts of the Chinese modal verbs and their translations. The English translations of "Yao" were identified and made into Table 4 for examination.

As can be seen from Table 4, the translation patterns of "Yao" demonstrate several trends. First, in every Chinese text, the high value modal verb was translated in English modal verbs of different value, with high and medium value ones occupying much larger proportions and low value ones smaller portions. Second, seen from the whole picture, among all the 1 717 occurrences of "Yao", 698 were translated into high-value English modal verbs, accounting for

**Table 3. Chinese Modal Verbs in 2000-2022 RWGs and their English Translations.**

|  | High Value | Percentage (%) | Medium Value | Percentage (%) | Low Value | Percentage (%) |
|---|---|---|---|---|---|---|
| Chinese Modal Verbs | 1981 | 84.84% | 162 | 6.94% | 192 | 8.22% |
| English Translation | 921 | 39.44% | 1136 | 48.65% | 278 | 11.9% |
| Differences | 1060 | 45.4% | -974 | -41.71% | -86 | -3.68% |

Note: for the convenience of operation, we classify the ellipsis, be + structures and others as low value (except that would and could are classified as medium value).

**Table 4. Classification of English Translations of "*Yao*" in 2000-2022 RWGs according to Value.**

| Year | Freq. of HVMV | Percentage (%) | Freq. of MVMV | Percentage (%) | Freq. of LVMV | Percentage (%) |
|------|------|------|------|------|------|------|
| 2000 | 26 | 23.42 | 79 | 71.17 | 6 | 5.40 |
| 2001 | 47 | 61.04 | 24 | 31.17 | 6 | 7.79 |
| 2002 | 37 | 49.33 | 34 | 45.33 | 4 | 5.33 |
| 2003 | 6 | 23.08 | 17 | 65.38 | 3 | 11.54 |
| 2004 | 55 | 49.55 | 50 | 45.05 | 6 | 5.40 |
| 2005 | 44 | 51.76 | 38 | 44.71 | 3 | 3.53 |
| 2006 | 61 | 61.00 | 35 | 35.00 | 4 | 4.00 |
| 2007 | 45 | 49.45 | 36 | 39.56 | 10 | 10.99 |
| 2008 | 44 | 45.36 | 50 | 51.55 | 3 | 3.09 |
| 2009 | 28 | 36.36 | 48 | 62.34 | 1 | 1.30 |
| 2010 | 33 | 34.74 | 59 | 62.11 | 3 | 3.16 |
| 2011 | 27 | 50.00 | 20 | 37.40 | 7 | 12.97 |
| 2012 | 8 | 17.78 | 35 | 77.78 | 2 | 4.44 |
| 2013 | 8 | 19.51 | 32 | 78.05 | 1 | 2.44 |
| 2014 | 25 | 35.21 | 46 | 64.79 | 0 | 0 |
| 2015 | 53 | 63.10 | 31 | 36.90 | 0 | 0 |
| 2016 | 16 | 27.59 | 37 | 63.79 | 5 | 8.62 |
| 2017 | 30 | 30.61 | 62 | 63.27 | 6 | 6.12 |
| 2018 | 24 | 33.80 | 44 | 61.97 | 3 | 4.23 |
| 2019 | 33 | 36.67 | 53 | 58.89 | 4 | 4.44 |
| 2020 | 20 | 33.33 | 30 | 50 | 10 | 16.67 |
| 2021 | 10 | 29.41 | 21 | 61.76 | 3 | 8.82 |
| 2022 | 18 | 27.28 | 47 | 71.21 | 1 | 1.51 |
| In total | 698 | 40.65 | 928 | 54.05 | 91 | 5.30 |

Note: HVMV stands for high value modal verbs, MVMV for medium value modal verbs, and LVMV for low value modal verbs.

40.65%; 928 were translated into medium-value English modal verbs, occupying for 53.18%; and 91 were translated into low-value ones, taking up 5.38%. Third, only 40.65% instances of "Yao" were translated into equivalent high value modal verbs, while near 60% occurrences of "Yao" were shifted into medium and low value ones. Such distribution of the English translation modal verbs shows that while translators maintain the high modal value in some instances, there is a clear tendency to moderate the intensity in the majority of cases, which suggests that the translators make conscious choices to adapt the text to the target culture's expectations and communicative norms.

The observed modal value shifts have several implications. First, by translating high-value modal verbs into medium and low-value equivalents, the translators create a more flexible and reader-friendly tone in the English versions of the Chinese RWGs [3,24]. This adjustment may make the Chinese RGWs more palatable to an international audience, as argued by Nida [1], which hence potentially reduce the psychological distance between the government and its readers. Besides, the shift from predominantly high-value modal verbs in Chinese to a more balanced distribution in English reflects an awareness of cultural differences in expressing authority and commitment. This adaptation demonstrates the translator's role as what Hatim & Mason termed as "a cultural mediator" [40]. In addition, as the literal meanings of modal verbs may change in different contexts and cultures, these shifts often aim to maintain pragmatic equivalence, ensuring that the intended effect of the original text is preserved in

the translation [7]. Last, as suggested by Halliday, the more varied use of modal values in the English translations may encourage greater reader engagement by presenting information in a less absolute manner, allowing for more nuanced interpretation [27].

In conclusion, the above analysis of modal verb translations in the 2000-2022 RWGs reveals a systematic approach by translators to mediate the psychological gap between the ST and TT. By modulating modal values, translators not only bridge linguistic differences but also adapt the text to meet the expectations and communication norms of the target audience. This strategic approach underscores the translator's critical role as a mediator in intercultural communication, particularly in the context of government reports where precise conveying of intent and policy is paramount.

## Discussion

The analysis of modal verb translation patterns reveals the complex role of translators as mediators between source and target texts. As Toury aptly points out, "any actual translation is necessarily a compromise between adequacy and acceptability" [41, p, 70], highlighting the delicate balance translators must maintain. This mediation process is particularly evident in the translation of Chinese government documents, where several factors influence translators' decisions.

First, the translators, in the translation process, need to consider the inherent nature of the Chinese RWGs as authoritative texts. As official and authoritative documents issued by the Chinese central government, they serve as crucial instruments for communicating China's policies and development strategies to the international community [42]. The translators, acting as a mediator, must preserve the authoritative nature of these documents while making them accessible to the target audience. This careful balancing act reflects what Hatim and Mason describe as the translator's role in negotiating between different discourse worlds [40].

As suggested by Tian, the prevalence of high-value modal verbs in the source text, particularly "Yao", demonstrates the government's strong commitment to its proposed actions and policies [24]. Following Halliday and Matthiessen's framework [43], these modal expressions in the source text convey both obligation and inclination, reflecting the government's determination and authority [22]. The translator's challenge lies in maintaining this authoritative tone while ensuring the translation aligns with diplomatic discourse conventions in English [44].

Second, the translators adopt mediation strategies to ensure the target text's acceptability and comprehensibility for international audiences. As Schäffner emphasizes, political translation requires careful consideration of the target audience's expectations and cultural norms [9]. This consideration is particularly evident in the translation of modal verbs, where the corpus analysis reveals a systematic modulation of modal values to meet the psychological expectations of the English speaking audiences.

The data shows that approximately 60% of high-value modal verbs in the source text are translated into medium or low-value equivalents in English, reflecting what House terms as "cultural filtering" [4] - a translation strategy that involves the translator's adjustment of linguistic and cultural elements to better suit the norms and expectations of the target audience. This strategic adjustment demonstrates translators' efforts to bridge the psychological gap between Chinese political discourse and international readers' expectations. For instance, the high-value modal verb "Yao" is frequently rendered as "will" or "should" rather than the more forceful "must," creating a more diplomatic and accessible tone for the international community. This modulation strategy aligns with what Li and Pan describe as "diplomatic recalibration" in political translation [10], where translators consciously adjust the rhetorical force to enhance cross-cultural acceptance while maintaining the essential message. Such mediation

reflects translators' crucial role in facilitating effective international communication by balancing source text authority with target audience receptivity.

Third, the translators, in the translation process, work as both translators and cultural mediators, a perspective that gained prominence with the "cultural turn" in translation studies. As Bassnett and Lefevere argue in their seminal work, translation operates as a form of cultural negotiation rather than mere linguistic transfer, where translators must navigate complex networks of cultural signs and values [45]. This cultural dimension of translation, they maintain, is fundamental to understanding how meaning is reconstructed across linguistic and cultural boundaries [46]. House further develops this perspective, emphasizing that translation is not merely a linguistic operation but a complex process of cultural mediation [4]. This is particularly evident in the translation of modal expressions between Chinese and English, where distinct cultural orientations shape linguistic preferences. As Bassnett notes, the translator's role as a cultural mediator involves making conscious choices that bridge not only linguistic but also cultural and ideological gaps between source and target texts, a process that requires deep understanding of both cultural systems and their respective communicative conventions [47].

Chinese political discourse, reflecting its collectivist cultural orientation, frequently employs high-value modal verbs combined with collective pronouns (e.g., "我们要" (we must), "我们必须"(we need)) to express strong commitment and shared responsibility [3,24,48]. In contrast, English political discourse, influenced by individualistic cultural values, tends to favor more hedged expressions and individual stance markers (e.g., "I believe," "I think", "it may be") to convey similar messages [9]. This cultural-linguistic difference requires translators to perform what Katan terms "cultural filtering," [8] adapting the modal expressions to align with target culture expectations while preserving the essential message.

The corpus analysis reveals that translators consistently mediate these cultural differences through strategic choices in modal verb translation. For instance, the high-value modal combinations in Chinese are often rendered into more moderated expressions in English, reflecting the pragmatic adaptation to achieve cultural appropriateness while maintaining communicative effectiveness [7].

## Conclusion

This corpus-based study of modal verb translations in the Chinese RWGs reveals significant insights into translators' role as mediators in bridging psychological gaps between source and target texts. The analysis demonstrates that translators systematically employ various strategies to navigate the complex interplay between linguistic accuracy and cultural appropriateness.

The findings highlight three key aspects of translators' mediation. First, translators consistently modulate high-value Chinese modals into medium or low-value English equivalents, reflecting what House terms "cultural filtering." [4] Second, they employ explicitation and implicitation strategies to balance source text fidelity with target text acceptability [7]. Third, translators demonstrate sophisticated awareness of cultural-linguistic differences in expressing authority and commitment, adapting their translations to align with international discourse conventions while maintaining the essential message of the source text.

These findings have important theoretical and practical implications. Theoretically, they contribute to our understanding of modality in political translation and underscore the crucial role of translators as cultural mediators. Practically, the insights offered can inform translation practices in diplomatic and political contexts, particularly in the translation of Chinese government documents for international audiences.

However, this study has certain limitations. The focus on modal verbs, while revealing, represents only one aspect of linguistic mediation. Future research could explore other linguistic features in political texts, examine the reception of translations by target audiences, or conduct comparative studies across different types of political documents. Additionally, investigating how translation strategies have evolved in response to changing international relations could provide valuable insights.

In conclusion, this study enhances our understanding of translators' mediating role in bridging psychological gaps between cultures through careful linguistic choices. The findings underscore the importance of viewing translation not merely as linguistic transfer but as a sophisticated form of cultural mediation that facilitates international communication.

## Supporting information

**S1 File. Chinese English Parallel Corpus of Chinese RWGs 2000-2022.**
(RAR)

## Author contributions

**Data curation:** Xujun Tian.

**Formal analysis:** Xujun Tian.

**Investigation:** Xujun Tian.

**Methodology:** Xujun Tian.

**Validation:** Xujun Tian.

**Writing – original draft:** Xujun Tian.

**Writing – review & editing:** Xujun Tian.

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
