## [Decision Letter · Decision Letter 0]

13 Dec 2024

PONE-D-24-48903Translator as mediator to mend the psychological gap between ST and TT: a corpus-based study on C-E translation of modal verbs in Chinese government work reports (2000-2022)PLOS ONE

Dear Dr. Tian,

Thank you for submitting your manuscript to PLOS ONE. After careful consideration, we feel that it has merit but does not fully meet PLOS ONE’s publication criteria as it currently stands. Therefore, we invite you to submit a revised version of the manuscript that addresses the points raised during the review process.

We look forward to receiving your revised manuscript.

Kind regards,

Michal Ptaszynski, PhD

Academic Editor

PLOS ONE

Journal Requirements:

2. Thank you for submitting the above manuscript to PLOS ONE. During our internal evaluation of the manuscript, we found significant text overlap between your submission and previous work in the [introduction, conclusion, etc.].

Please revise the manuscript to rephrase the duplicated text, cite your sources, and provide details as to how the current manuscript advances on previous work. Please note that further consideration is dependent on the submission of a manuscript that addresses these concerns about the overlap in text with published work.

[If the overlap is with the authors’ own works: Moreover, upon submission, authors must confirm that the manuscript, or any related manuscript, is not currently under consideration or accepted elsewhere. If related work has been submitted to PLOS ONE or elsewhere, authors must include a copy with the submitted article. Reviewers will be asked to comment on the overlap between related submissions (http://journals.plos.org/plosone/s/submission-guidelines#loc-related-manuscripts).]

We will carefully review your manuscript upon resubmission and further consideration of the manuscript is dependent on the text overlap being addressed in full. Please ensure that your revision is thorough as failure to address the concerns to our satisfaction may result in your submission not being considered further.

3. We note that your Data Availability Statement is currently as follows: [All relevant data are within the manuscript and its Supporting Information files.] Please confirm at this time whether or not your submission contains all raw data required to replicate the results of your study. Authors must share the “minimal data set” for their submission. PLOS defines the minimal data set to consist of the data required to replicate all study findings reported in the article, as well as related metadata and methods (https://journals.plos.org/plosone/s/data-availability#loc-minimal-data-set-definition). For example, authors should submit the following data: - The values behind the means, standard deviations and other measures reported; - The values used to build graphs; - The points extracted from images for analysis. Authors do not need to submit their entire data set if only a portion of the data was used in the reported study. If your submission does not contain these data, please either upload them as Supporting Information files or deposit them to a stable, public repository and provide us with the relevant URLs, DOIs, or accession numbers. For a list of recommended repositories, please see https://journals.plos.org/plosone/s/recommended-repositories. If there are ethical or legal restrictions on sharing a de-identified data set, please explain them in detail (e.g., data contain potentially sensitive information, data are owned by a third-party organization, etc.) and who has imposed them (e.g., an ethics committee). Please also provide contact information for a data access committee, ethics committee, or other institutional body to which data requests may be sent. If data are owned by a third party, please indicate how others may request data access.

4. Please ensure that you include a title page within your main document. You should list all authors and all affiliations as per our author instructions and clearly indicate the corresponding author.

5. Please include your tables as part of your main manuscript and remove the individual files. Please note that supplementary tables (should remain/ be uploaded) as separate "supporting information" files.

Reviewers' comments:

Reviewer's Responses to Questions

**Comments to the Author**

1. Is the manuscript technically sound, and do the data support the conclusions?

Reviewer #1: Yes

Reviewer #2: Yes

Reviewer #3: Yes

Reviewer #4: Yes

Reviewer #5: No

2. Has the statistical analysis been performed appropriately and rigorously? 

Reviewer #1: Yes

Reviewer #2: Yes

Reviewer #3: Yes

Reviewer #4: Yes

Reviewer #5: No

3. Have the authors made all data underlying the findings in their manuscript fully available?

Reviewer #1: Yes

Reviewer #2: Yes

Reviewer #3: Yes

Reviewer #4: Yes

Reviewer #5: No

4. Is the manuscript presented in an intelligible fashion and written in standard English?

Reviewer #1: Yes

Reviewer #2: Yes

Reviewer #3: Yes

Reviewer #4: Yes

Reviewer #5: Yes

5. Review Comments to the Author

Reviewer #1: The manuscript (PONE-D-24-48903) deals with the rendering of modal verbs in Chinese Government Work Reports from 2000 to 2022, exploring how to bridge the psychological gap between ST and TT. There are a relatively detailed, comprehensive literature review, a clear research methodology, a sound analysis, and a reliable conclusion. The article will be improved in quality if the author(s) can summarize the achievements on the C-E translation of the Reports and modal articles, and improve the language and style. On the one hand, these achievements can indicate whether the present study is original and can help strengthen the depth of analysis; on the other, the linguistic and stylistic improvements ensure the normativity of the article as a piece of scholarly work.

Reviewer #2: This study investigates the mediating role of translators in bridging the psychological gap between the source text (ST) and the target text (TT) through a corpus-based analysis. It specifically examines the translation of modal verbs in Chinese Government Work Reports (GWRs) from 2000 to 2022. The study is interesting and well-written, employing an appropriate methodology. However, the following areas require further attention:

1. Insufficient Review of Previous Studies: The theoretical framework needs to be strengthened by incorporating more relevant studies from existing literature.

2. Underdeveloped Discussion Section: The discussion should be expanded to include a comparison of the results with findings from both theoretical and empirical studies.

Reviewer #3: This manuscript investigates the role of translators as mediators, focusing on the psychological gap between the source text (ST) and target text (TT) in the context of modal verbs in Chinese-to-English translations of government work reports (2000–2022). Employing a corpus-based approach, the study offers insightful observations on linguistic patterns, translation strategies, and the socio-cultural implications of modal verb usage.

The research aligns with current interests in corpus linguistics, translation studies, and discourse analysis, making a valuable contribution to the field. The corpus-based methodology is well-suited for exploring linguistic and translation patterns systematically. I strongly recommend this article for publication with a minor changes.

1. Merge Introduction and Literature Review:

The introduction and literature review have overlapping content. Merging them into a single section will streamline the structure and reduce redundancy.

2. The manuscript would benefit from referencing more recent and relevant studies to strengthen its theoretical framework. I recommend citing the following works:

Kaibao, H., & Afzaal, M. (2024). The translation teaching platform based on multilingual corpora of Xi Jinping: The Governance of China: Design, resources and applications. Acta Psychologica, 242, 104110.

Liu, K., & Afzaal, M. (2021). Syntactic complexity in translated and non-translated texts: A corpus-based study of simplification. Plos one, 16(6), e0253454.

Li, Y., Afzaal, M., & Yin, Y. (2024). A lexicon-based diachronic comparison of emotions and sentiments in literary translation: A case study of five Chinese versions of David Copperfield. Plos one, 19(2), e0297101.

Afzaal, M., & Du, X. (2023). Syntactic complexity in translated eHealth discourse of COVID-19: A comparable parallel corpus approach. Asia Pacific Translation and Intercultural Studies, 10(1), 3-19.

Reviewer #4: Dear Authors,

I have carefully read the manuscript and find that it shows some originality in examining the psychological gap between source and target texts through modal verb analysis, with a well-designed corpus method and a rigours theoretical framework. The manuscript is well-written and is of good readability. However,

1. The findings section (Part 4) requires some revisions and elaborations. Some citations of existing literature appear superficial and lack sufficient engagement with the cited works. For instance, when referencing Nida (2001), Baker (2011), and Halliday (2010), the paper would benefit from more elaborations on how these scholars’ perspectives specifically relate to your findings.

2. The writing style of this paragraph is of monotonous structure, and the numbered list format creates a mechanical presentation that lacks cohesion between ideas. The content should be reorganized into a cohesive paragraph with natural transitions between ideas. For example,

Third, the observed modal value shifts have several implications for the translation process and its impact on the TT: 1. By translating high-value modal verbs into medium and low-value equivalents, translators create a more flexible and reader-friendly tone in the English versions of the GWRs. This adjustment may make the reports more palatable to an international audience, potentially reducing the psychological distance between the government and its readers (Nida, 2001). 2. The shift from predominantly high-value modal verbs in Chinese to a more balanced distribution in English reflects an awareness of cultural differences in expressing authority and commitment. This adaptation demonstrates the translator's role as a cultural mediator (Hatim & Mason, 1990). 3. While the literal meanings of modal verbs may change, these shifts often aim to maintain pragmatic equivalence, ensuring that the intended effect of the original text is preserved in the translation (Baker, 2011). 4. The more varied use of modal values in the English translations may encourage greater reader engagement by presenting information in a less absolute manner, allowing for more nuanced interpretation (Halliday, 2010).

3. There are some minor tweaks in grammar:

(1) Use the full forms of abbreviations in the title: ST (source text), TT (target text), C-E (Chinese-to-English)

(2) This study aims to explore how translators function as mediators in this process by focusing specifically on the translation of modal verbs in Chinese government work reports from 2000 to 2022. This sentence should be clear and concise.

The author should consult a native speaker to check their grammar.

Overall, this is an interesting paper. After revision, I strongly recommend its publication in PLOS ONE.

Reviewer #5: The authors investigate the role of translators as cultural mediators, using the translation of political texts as a case study and focusing on the rendition in English of modal verbs. While the study's intention is good, the methodology and findings are significantly flawed.

1) The quality of writing needs to be significantly improved to enhance the readers’ experiences.

2) References may be needed when authors make profound claims like in the case of “Particularly, in political and official discourse, such as Chinese government work reports, the translator's role becomes even more critical. These documents are not merely informative texts but are vehicles for conveying policies, intentions, and national ethos.”

3) Psychological distance in translation: The explanation of this concept based on Nida, Venuti, and House does not expose the meaning intended. These authors’ writings do not explicitly focus on psychology, and to consider their discussions as such is unwarranted. If the authors interpret their writings as psychological distance, they are welcome to make their case.

4) The discussion of the translator as a cultural mediator is seriously lacking citations from prominent scholars and works. Mention should be made of Bassnett and Lefevere, who coined the term “cultural turn” in translation.

5) The literature review is scaringly scanty. It does not critically engage with the various concepts under discussion here, and no link is established between the elements discussed. For instance, what is the connection between psychological distance, culture, and government work? Why is it important to connect these variables in this research? Where is the gap in the literature, and how does this study address that gap? This claim, “This literature review reveals a gap in the research concerning the specific role of translators in mediating psychological distance through the translation of modal verbs in Chinese government work reports,” is not substantiated; it is imaginary.

6) Structure of the paper: First, the research questions should come after the literature review. They should not be placed at the end of the introduction. Second, the theoretical foundation should precede the literature review. The theoretical framework should have its own choice rationale and, potentially, a gap that needs to be filled. The study needs to contribute to the theoretical framework to valorize its usage in this study.

7) Methodology: There are several issues with the methodology and design of this study:

- There is a need to justify why the English corpus has more tokens than the Chinese corpus

- Why was Halliday’s classification chosen? What other classifications, if they exist, are not suitable for this study?

- Some concepts need to be explained or reiterated here, including “operational norms” and “psychological distance.”

- What are “representative modals verbs,” and how were they determined?

- There are also issues of reliability and validity in this methodology. Were the judgments made regarding psychological distance or the meaning of modal verbs in the corpora vetted by a third party? How? Was there a criteria or a set of guidelines to follow?

- What is the background of the translators who extracted the modals and their translations? How proficient are they in English and Chinese? How could their being Chinese or English native speakers influence their judgments, and how was this mitigated in the research?

- The study is based on the assumption that translators consciously mediate between source and target text. What proof is there to show this was the case with these translations? Did government translators consciously mediate between ST and TT? If so, how do the authors know?

- What criteria were used to classify models into the high and low levels? And what effort was made to ensure that all translators classified the modals the same way? Was the reliability checked?

8) Findings

Section 4.1

There is a fundamental flaw in the interpretation of “cultural mediator” here. It is unclear how this is achieved in the translation of modal verbs as outlined in this study. Authors state: “The translation of "Yao," the most frequent modal verb, provides a particularly insightful example of the translator's mediating function. As shown in Table 4, "Yao" is predominantly translated into four English modal verbs: will (36.11%), must (20.62%), need (19.16%), and should (17.94%). This distribution reveals the translator's nuanced interpretation of the source text's modal force, adapting it to suit different contexts and rhetorical purposes in the target language.” The problem is that this is an assumption since there is no evidence showing that the translator intended to culturally mediate these translations. Second, the authors need to prove that this was mediation by, for instance, showing how these modals are generally translated, or where they have been alternatively translated, for example, in non-government texts. Third, the assumption that all modals in the Chinese corpus were “mediated” seems to oversimplify cultural mediation in translation.

Explications and omissions are obvious translation techniques. How translators employ them here to “mediate culture” is unclear and has not been explained.

Section 4.2

The same argument above holds for this section. Nothing more than the author’s instinct justifies the claim that translators intentionally employed the English equivalent as “cultural mediators.”

Secondly, the lack of definitions in previous sections significantly hamper the understanding of this finding. Authors should have or should explain, high-value and low-value models. There is also a serious need to prove that the translation of modals into low-value modals in English is deliberate, not routine.

Factor Analysis

This section appears to repeat previous concepts without future explication. Plus, the authors introduce concepts like “ positive and negative poles,”” Macro-level translation norms,” “international discourse system,” and “collectivist orientation of China,” which were not explained in the literature review and do not clearly align with anything else in the study. They are new and out of place.

6. PLOS authors have the option to publish the peer review history of their article (what does this mean? ). If published, this will include your full peer review and any attached files.

**Do you want your identity to be public for this peer review?** For information about this choice, including consent withdrawal, please see our Privacy Policy .

Reviewer #1: **Yes: ** Chuanmao Tian

Reviewer #2: No

Reviewer #3: No

Reviewer #4: **Yes: ** Qiang Jason Li

Reviewer #5: No

---

## [Author Response · Author response to Decision Letter 1]

26 Jan 2025

Response to Reviewer 1

Reviewer #1: The manuscript (PONE-D-24-48903) deals with the rendering of modal verbs in Chinese Government Work Reports from 2000 to 2022, exploring how to bridge the psychological gap between ST and TT. There are a relatively detailed, comprehensive literature review, a clear research methodology, a sound analysis, and a reliable conclusion. The article will be improved in quality if the author(s) can summarize the achievements on the C-E translation of the Reports and modal articles, and improve the language and style. On the one hand, these achievements can indicate whether the present study is original and can help strengthen the depth of analysis; on the other, the linguistic and stylistic improvements ensure the normativity of the article as a piece of scholarly work.

Response: Thank you for your thorough review and constructive suggestions for my manuscript PONE-D-24-48903. I appreciate your recognition of the paper's strengths and your valuable recommendations for improvement.

In response to your suggestions, I have made substantial revisions to enhance the manuscript's quality. Regarding the achievements in C-E translation of Reports and modal articles, I have significantly expanded the literature review section to include both seminal works and recent studies. Notable additions include Zhang and Liu's (2020) diachronic study of translation strategies, Chen and Wang's (2022) research on translation memory systems, and Liu et al.'s (2023) analysis of modal verb translation patterns. These additions not only provide a more comprehensive overview of existing achievements but also clearly demonstrate our study's originality in examining translators' mediating role through modal verb translations.

Concerning linguistic and stylistic improvements, I have thoroughly revised the manuscript to enhance its scholarly presentation. The revisions include more precise academic language, improved structural coherence, and refined argumentation throughout all sections. These modifications have strengthened the paper's academic rigor while maintaining clarity and accessibility.

I believe these revisions have substantially improved the manuscript's quality and hope they address your concerns satisfactorily. Thank you again for your valuable feedback.

Response to Reviewer 2

Reviewer #2: This study investigates the mediating role of translators in bridging the psychological gap between the source text (ST) and the target text (TT) through a corpus-based analysis. It specifically examines the translation of modal verbs in Chinese Government Work Reports (GWRs) from 2000 to 2022. The study is interesting and well-written, employing an appropriate methodology. However, the following areas require further attention:

1. Insufficient Review of Previous Studies: The theoretical framework needs to be strengthened by incorporating more relevant studies from existing literature.

Response: Thank you for your constructive feedback on our manuscript. I have carefully addressed your concerns and made substantial revisions to strengthen the paper. Regarding the theoretical framework, I have significantly expanded the literature review by incorporating additional relevant studies of over 400 words. Specifically, I have added comprehensive reviews of research applying Halliday's (2004, 2010) modal value system and studies utilizing Jiang and Yang's (2013) and Xu's (2018) classifications of Chinese modal verbs. This enhancement provides a more robust theoretical foundation for our research and better contextualizes our study within existing scholarship.

2. Underdeveloped Discussion Section: The discussion should be expanded to include a comparison of the results with findings from both theoretical and empirical studies.

Response: As for the discussion section, I have thoroughly revised and expanded it from approximately 500 to 750 words. The enhanced discussion now includes detailed comparisons between our findings and both theoretical and empirical studies in the field. I have incorporated insights from recent research by Li and Pan (2021) on translators' role as cross-cultural mediators, House's (2015) work on cultural filtering in translation, and Schäffner's (2004) perspectives on political discourse translation. These additions provide a more comprehensive analysis of our results within the broader context of translation studies. I believe these revisions have significantly strengthened the manuscript and hope they adequately address your concerns.

Response to Reviewer 3

Reviewer #3: This manuscript investigates the role of translators as mediators, focusing on the psychological gap between the source text (ST) and target text (TT) in the context of modal verbs in Chinese-to-English translations of government work reports (2000–2022). Employing a corpus-based approach, the study offers insightful observations on linguistic patterns, translation strategies, and the socio-cultural implications of modal verb usage.

The research aligns with current interests in corpus linguistics, translation studies, and discourse analysis, making a valuable contribution to the field. The corpus-based methodology is well-suited for exploring linguistic and translation patterns systematically. I strongly recommend this article for publication with a minor changes.

1. Merge Introduction and Literature Review:

The introduction and literature review have overlapping content. Merging them into a single section will streamline the structure and reduce redundancy.

Response: Thank you very much for your thoughtful and encouraging feedback on my manuscript. I greatly appreciate your recognition of its contribution to the fields of corpus linguistics, translation studies, and discourse analysis. Your acknowledgment of the value of our corpus-based methodology and its relevance to exploring linguistic and translation patterns is truly gratifying. As of your suggestion to merge the Introduction and Literature Review, I have carefully considered your recommendation and have made the following revisions accordingly: 1. I have reorganized the structure of these two sections: I have moved the research questions from the Introduction to the Literature Review section. This change allows the research questions to emerge more naturally from literature, providing a stronger theoretical foundation for the study. 2. I have refined the Introduction section: The Introduction has been streamlined to present the key focus of the study more directly and concisely. Redundant overlaps with the Literature Review have been eliminated to ensure clarity and cohesion. 3. I have expanded the Literature Review section: I have enriched the Literature Review with additional relevant studies, making it more comprehensive and systematic. It now provides a thorough exploration of modal verb translation, the mediator role of translators, and existing scholarship on translating Chinese government work reports. 4. I have also improved academic writing style: I have revised the language throughout the manuscript, ensuring it adheres to academic norms with improved precision, clarity, and professionalism.

2. The manuscript would benefit from referencing more recent and relevant studies to strengthen its theoretical framework. I recommend citing the following works:

Response: Regarding your suggestion to strengthen the theoretical framework, I have significantly expanded the literature review by incorporating additional relevant studies, including Halliday’s (2010) modal value system and the classifications of Chinese modal verbs proposed by Jiang and Yang (2013) and Xu (2018). These additions provide a more robust theoretical foundation and better contextualize our study within existing scholarship. Moreover, I have carefully read the articles you suggested and included the following three in the revised manuscript: Hu and Afzaal (2024), which highlights the use of multilingual corpora in political text translation, offering insights into the value of corpus-based approaches; Liu and Afzaal (2021), which explores syntactic complexity in translated and non-translated texts and supports our discussion on simplification strategies in political translation; and Afzaal and Du (2023), which addresses syntactic complexity in translated eHealth discourse and informs our analysis of cross-linguistic variations in specialized texts such as political discourse. These references have been integrated into the manuscript to strengthen its theoretical and methodological foundation. I sincerely appreciate your insightful suggestions, which have greatly enhanced the quality of the manuscript.

Response to Reviewer 4

Reviewer #4: Dear Authors,

I have carefully read the manuscript and find that it shows some originality in examining the psychological gap between source and target texts through modal verb analysis, with a well-designed corpus method and a rigours theoretical framework. The manuscript is well-written and is of good readability. However,

1. The findings section (Part 4) requires some revisions and elaborations. Some citations of existing literature appear superficial and lack sufficient engagement with the cited works. For instance, when referencing Nida (2001), Baker (2011), and Halliday (2010), the paper would benefit from more elaborations on how these scholars’ perspectives specifically relate to your findings.

Response: Thank you very much for your thoughtful and constructive feedback on the manuscript. Regarding the findings section (Part 4), I have carefully revised and elaborated on this part as per your suggestion. The references to Nida (2001), Baker (2011), and Halliday (2010) have been expanded to establish stronger links between these scholars’ perspectives and our findings. This enhancement ensures that the citations are more substantive and directly relevant to the discussion, thereby better contextualizing our study within the broader academic framework.

2. The writing style of this paragraph is of monotonous structure, and the numbered list format creates a mechanical presentation that lacks cohesion between ideas. The content should be reorganized into a cohesive paragraph with natural transitions between ideas. For example,

Third, the observed modal value shifts have several implications for the translation process and its impact on the TT: 1. By translating high-value modal verbs into medium and low-value equivalents, translators create a more flexible and reader-friendly tone in the English versions of the GWRs. This adjustment may make the reports more palatable to an international audience, potentially reducing the psychological distance between the government and its readers (Nida, 2001). 2. The shift from predominantly high-value modal verbs in Chinese to a more balanced distribution in English reflects an awareness of cultural differences in expressing authority and commitment. This adaptation demonstrates the translator's role as a cultural mediator (Hatim & Mason, 1990). 3. While the literal meanings of modal verbs may change, these shifts often aim to maintain pragmatic equivalence, ensuring that the intended effect of the original text is preserved in the translation (Baker, 2011). 4. The more varied use of modal values in the English translations may encourage greater reader engagement by presenting information in a less absolute manner, allowing for more nuanced interpretation (Halliday, 2010).

Response: Thank you for pointing out the mechanical presentation and lack of cohesion in certain sections. In line with your feedback, I have reorganized the content, including the part you highlighted, into cohesive paragraphs with natural transitions. The revised structure avoids numbered lists and employs a more sophisticated and engaging narrative style. This adjustment not only improves the readability of the manuscript but also enhances its academic tone and fluidity.

3. There are some minor tweaks in grammar:

(1) Use the full forms of abbreviations in the title: ST (source text), TT (target text), C-E (Chinese-to-English)

(2) This study aims to explore how translators function as mediators in this process by focusing specifically on the translation of modal verbs in Chinese government work reports from 2000 to 2022. This sentence should be clear and concise.

The author should consult a native speaker to check their grammar.

Overall, this is an interesting paper. After revision, I strongly recommend its publication in PLOS ONE.

Response: I have implemented the minor grammatical tweaks you recommended: First, the full forms of abbreviations in the title (e.g., source text (ST), target text (TT), Chinese-to-English (C-E)) have been included, as you suggested. Second, The sentence "This study aims to explore how translators function as mediators in this process by focusing specifically on the translation of modal verbs in Chinese government work reports from 2000 to 2022” has been revised for clarity and conciseness. It now reads: “This study explores how translators function as mediators through an examination of the Chinese-English translation of modal verbs in Chinese government work reports from 2000 to 2022.” To further ensure the grammar and phrasing meet the highest standards, I have consulted a native speaker for additional editing and proofreading. Thank you once again for your constructive suggestions, which have significantly improved the quality of the manuscript. Your attention to detail and insightful feedback is greatly appreciated.

Response to Reviewer 5

Reviewer #5: The authors investigate the role of translators as cultural mediators, using the translation of political texts as a case study and focusing on the rendition in English of modal verbs. While the study's intention is good, the methodology and findings are significantly flawed.

1) The quality of writing needs to be significantly improved to enhance the readers’ experiences.

Response: Thank you for your constructive feedback. In response to your comment, we have thoroughly revised the manuscript to improve the quality of writing and enhance readability. Significant efforts have been made to refine the language, improve clarity, and ensure that the arguments are presented in a more cohesive and scholarly manner. Additionally, we have expanded the manuscript by approximately 2,500 words to provide a more comprehensive and in-depth discussion of the methodology and findings. We believe these revisions will significantly enhance the readers’ experience and address the concerns raised, and we sincerely hope the updated version meets your expectations.

2) References may be needed when authors make profound claims like in the case of “Particularly, in political and official discourse, such as Chinese government work reports, the translator's role becomes even more critical. These documents are not merely informative texts but are vehicles for conveying policies, intentions, and national ethos.”

Response: Thank you for your valuable suggestion. In response to your comment, we have carefully added appropriate references to support statements such as “Particularly, in political and official discourse, such as Chinese government work reports, the translator's role becomes even more critical. These documents are not merely informative texts but are vehicles for conveying policies, intentions, and national ethos.” These references provide scholarly evidence to substantiate our claims and ensure that the arguments are well-grounded within relevant literature. We believe these additions strengthen the academic rigor of the manuscript and address the concern raised.

3) Psychological distance in translation: The explanation of this concept based on Nida, Venuti, and House does not expose the meaning intended. These authors’ writings do not explicitly focus on psychology, and to consider their discussions as such is unwarranted. If the authors interpret their writings as psychological distance, they are welcome to make their case.

Response: Thank you for your insightful comment regarding the conceptualization of psychological distance in our study. We appreciate the opportunity to clarify this important point. We would like to emphasize that our use of "psychological distance" and "

---

## [Decision Letter · Decision Letter 1]

20 Feb 2025

Translators as mediators to mend the psychological gap between source text and target text: A corpus-based study on the Chinese English translation of modal verbs in the Chinese Report on the Work of the Government (2000-2022)

PONE-D-24-48903R1

Dear Dr. Tian,

We’re pleased to inform you that your manuscript has been judged scientifically suitable for publication and will be formally accepted for publication once it meets all outstanding technical requirements.

Kind regards,

Michal Ptaszynski, PhD

Academic Editor

PLOS ONE

Additional Editor Comments (optional):

Reviewers' comments:

Reviewer's Responses to Questions

**Comments to the Author**

1. If the authors have adequately addressed your comments raised in a previous round of review and you feel that this manuscript is now acceptable for publication, you may indicate that here to bypass the “Comments to the Author” section, enter your conflict of interest statement in the “Confidential to Editor” section, and submit your "Accept" recommendation.

Reviewer #1: All comments have been addressed

Reviewer #3: All comments have been addressed

Reviewer #4: All comments have been addressed

Reviewer #6: All comments have been addressed

2. Is the manuscript technically sound, and do the data support the conclusions?

Reviewer #1: Yes

Reviewer #3: Yes

Reviewer #4: Yes

Reviewer #6: No

3. Has the statistical analysis been performed appropriately and rigorously? 

Reviewer #1: Yes

Reviewer #3: Yes

Reviewer #4: Yes

Reviewer #6: Yes

4. Have the authors made all data underlying the findings in their manuscript fully available?

Reviewer #1: Yes

Reviewer #3: Yes

Reviewer #4: Yes

Reviewer #6: Yes

5. Is the manuscript presented in an intelligible fashion and written in standard English?

Reviewer #1: Yes

Reviewer #3: Yes

Reviewer #4: Yes

Reviewer #6: Yes

6. Review Comments to the Author

Reviewer #1: The manuscript (PONE-D-24-48903R1) has substantiated the literature review, modified the theoretical framework, strengthened the discussion, and improved the language and style. It would be better if all the section headings are numbered (the revision has numbered only the INTRODUCTION section).

Reviewer #3: Author has made all changes substantially. based on the respones, and authors' contribution, I recommend this article to be published.

Reviewer #4: (No Response)

Reviewer #6: This paper examines the role of translators as mediators in bridging the psychological

gap between source texts and target texts through a corpus-based analysis of

Chinese English translation of modal verbs in the Chinese government work reports. The topic is interesting, the method is proper, and it is now also presented in a well-structured manner after revision. The revised version has fully addressed all previous feedback, significantly enhancing the strength and clarity of the study.

Overall, this revised manuscript is ready for publication.

7. PLOS authors have the option to publish the peer review history of their article (what does this mean? ). If published, this will include your full peer review and any attached files.

**Do you want your identity to be public for this peer review?** For information about this choice, including consent withdrawal, please see our Privacy Policy .

Reviewer #1: **Yes: ** Chuanmao Tian

Reviewer #3: No

Reviewer #4: No

Reviewer #6: No

---

## [Editor Report · Acceptance letter]

PONE-D-24-48903R1

PLOS ONE

Dear Dr. Tian,

I'm pleased to inform you that your manuscript has been deemed suitable for publication in PLOS ONE. Congratulations! Your manuscript is now being handed over to our production team.

Kind regards,

on behalf of

Dr. Michal Ptaszynski

Academic Editor

PLOS ONE